# Guiding Reasoning in Small Language Models with LLM Assistance

## Abstract

Small language models (SLMs) typically falter on tasks requiring deep, multi-step reasoning. This paper introduces SMART (Small Reasons, Large Hints), a framework where large language models (LLMs) provide targeted, selective guidance to augment SLM reasoning. Drawing from cognitive scaffolding, SMART uses a score-based mechanism to identify uncertain SLM reasoning steps, triggering LLM correction only when essential. This approach, framing structured reasoning as an optimal policy search, steers SLMs towards correct solutions without exhaustive sampling. On mathematical reasoning datasets, SMART enables SLMs to achieve up to 98.9% of LLM-level performance while reducing LLM token usage by up to 90.0%. Our work paves the way for collaborative use of both SLM and LLM to tackle complex reasoning tasks that are currently unsolvable by SLMs alone.

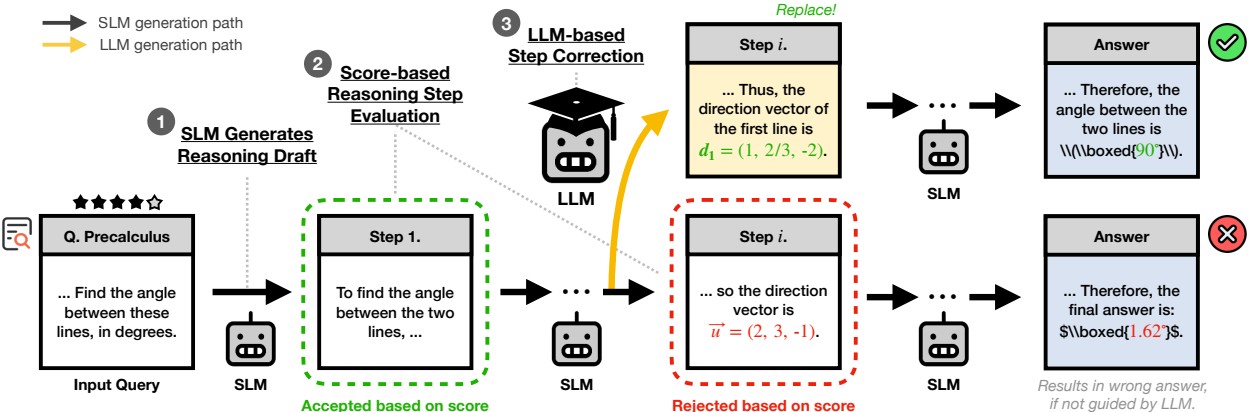

Figure 1: Overview of our SMART framework. First, the SLM generates an initial reasoning trajectory, producing step-by-step solutions. Second, after each step is generated, a score-based evaluation mechanism assesses its reliability, determining whether it meets a predefined threshold. Third, for the steps identified as uncertain, the LLM is queried to generate new step, replacing the original SLM step while preserving the preceding SLM-generated steps, and then the process continues until the EOS token or the final answer is reached.

## 1 Introduction

Large language models (LLMs) have demonstrated powerful reasoning capabilities, extending their impact beyond NLP to fields like robotics (Driess et al., 2023) and fundamental science (Madani et al., 2023; Thirunavukarasu et al., 2023). Their ability to perform multi-step logical deductions and structured decision-making is often attributed to System 2-like reasoning—deliberate, step-by-step cognitive processes analogous to human analytical thought (Kahneman, 2011; Guan et al., 2024). Unlike heuristic-driven System 1 reasoning, this System 2-like deliberation is crucial for complex tasks such as mathematical problem-solving, scientific

reasoning, and strategic planning (Snell et al., 2024). Recent studies suggest that LLMs, when properly prompted or trained via reinforcement learning (RL), can exhibit emergent System 2-like behavior, proving effective for structured reasoning (Shao et al., 2024; Guo et al., 2025).

However, small language models (SLMs), despite their efficiency, struggle with such structured reasoning due to their limited capacity and lack of emergent cognitive patterns (Mirzadeh et al., 2024). Their reasoning is often shallow, relying more on memorization and surface-level heuristics rather than deep logical deduction (Nikankin et al., 2024). This limitation raises a critical question:

> **Research question**
>
> If SLMs inherently lack reasoning capabilities, does this mean they are unusable for settings demanding complex reasoning?

To address this challenge, we introduce a reasoning framework inspired by *scaffolding*, a concept from cognitive science that describes how humans rely on external support—such as guidance from a teacher or structured tools—to complete tasks beyond their independent capability (Wellman & Gelman, 1992; Battaglia et al., 2013; Gerstenberg & Stephan, 2021). Analogously, in our framework, an SLM acts as a primary reasoner while selectively integrating LLM-generated reasoning steps only when it encounters unreliable states (Lake et al., 2017). Adaptivity is achieved via a score-based mechanism that determines when LLM guidance is necessary. At each step, the SLM's candidate reasoning is scored (e.g., using a Process Reward Model (PRM) or averaged token-level confidence over the sequence). If the score is below a threshold, an LLM generates a replacement step (Figure 1); otherwise, the SLM proceeds independently. This ensures LLM intervention only at critical junctures, reducing costs and improving reasoning robustness.

A particularly striking observation in our study is that while increasing the number of sampled reasoning trajectories (e.g., Best-of-N sampling) naturally improves performance, this approach is computationally expensive and often lacks principled guidance—frequently relying on lucky guesses rather than a structured search for optimal reasoning (Snell et al., 2024). In contrast, our framework achieves more significant improvements while modifying only specific reasoning steps rather than re-evaluating the entire trajectory. This suggests that rather than globally increasing sampling and scaling compute, selectively correcting key decisions is sufficient to guide the reasoning process toward an optimal solution (Sharma et al., 2023).

From an RL perspective, structured reasoning can be viewed as an optimal policy search, where the model must navigate a sequence of states to reach a correct answer (Chen et al., 2024a). Our findings indicate that even when an SLM struggles to determine the next step in a reasoning trajectory, providing guidance only at critical points is enough to keep the overall trajectory on an optimal path. This selective scaffolding is non-trivial; while standard RL methods like diffusion models (Ren et al., 2024) or policy optimization (Cetin & Celiktutan, 2022) might seem applicable, they often demand extensive exploration or dense rewards, proving computationally prohibitive (Ding et al., 2024). Our results indicate that even without full trajectory optimization, targeted scaffolding can provide sufficient correction, suggesting a promising direction for future integration with RL-based structured reasoning frameworks. Our key contributions are:

- We introduce a novel framework called Small Reasons, Large Hints (`SMART`), where an SLM reasons but selectively incorporates LLM-generated reasoning steps (Section 2).

- We conduct experiments on mathematical reasoning benchmark, demonstrating that SLM reasoning, selectively incorporating response from LLM scaffolding, can reach up to **98.9%** of LLM accuracy while reducing up to **90.0%** of LLM token usage. (Section 3).

- We analyze `SMART`'s guided reasoning, demonstrating critical LLM intervention at initial steps and `SMART`'s highly concise outputs for incorrect responses, offering insights into hybrid reasoning system (Section 4).

Table 1: Relative performance of SMART compared to LLM performance according to the difficulty level of problem on the MATH500 dataset. Each cell shows the percentage of our performance to that of the target LLM (i.e., 100% means SMART brings the identical performance with that of LLM). We show $N$=16 for all settings. The values in parentheses indicate the improvement of SMART over the SLM baseline, with colors denoting the direction of change. Performance improvement larger than 20% are bold-faced.

| Model | Search | Lv 1 | Lv 2 | Lv 3 | Lv 4 | Lv 5 |
|---|---|---|---|---|---|---|
| Qwen 1.5B-7B | Best-of-N | 100.00% (+5.20%) | 98.65% (+5.14%) | 98.56% (+18.88%) | 98.61% (**+24.39%**) | 99.12%(**+73.20%**) |
| | Beam Search | 100.00% (+2.53%) | 94.96% (+2.76%) | 92.44% (+7.48%) | 87.33% (+13.88%) | 92.70% (**+46.42%**) |
| Llama 1B-8B | Best-of-N | 102.60% (+14.66%) | 101.42% (+15.05%) | 98.86% (**+38.97%**) | 97.12% (**+37.85%**) | 94.94% (**+141.55%**) |
| | Beam Search | 92.28% (+0.00%) | 97.36% (+14.06%) | 88.99% (+8.93%) | 90.09% (**+23.15%**) | 114.57% (**+39.23%**) |

## 2 SMART

### 2.1 Preliminaries and notations

We formalize the reasoning process as an iterative decision-making problem. Given a query $Q$, a reasoning trajectory $R = (r_1, r_2, \ldots, r_m)$ consists of a sequence of intermediate step $r_i$, leading to a final answer $A$. The probability of generating $R$ given $Q$ is modeled as:

$$P(R \mid Q) = \prod_{i=1}^{m} P(r_i \mid Q, r_{<i}). \tag{1}$$

The answer $A$ is then determined as:

$$A = \arg\max_a P(a \mid Q, R). \tag{2}$$

In this formulation, each $r_i$ represents an intermediate reasoning step, and the correctness of the final answer depends on the entire trajectory $R$.

### 2.2 Motivation

SLMs are efficient but frequently fail to generate globally coherent reasoning trajectories due to their limited capacity. A straightforward way to improve reasoning performance is to increase test-time compute (Snell et al., 2024), such as generating multiple trajectories via Best-of-N sampling and selecting the most probable path: $R^* = \arg\max_{R^{(j)}} P(R^{(j)} \mid Q)$ where $R^{(j)} \sim P_{\text{SLM}}(R \mid Q)$. However, such approaches introduce significant computational overhead, requiring exponentially more sampling as reasoning complexity increases. Additionally, such selection does not inherently optimize for logical correctness; instead, it favors heuristic shortcuts that may align with fluency but not necessarily with accurate reasoning.

One motivation for our approach is the observation that not all reasoning steps for a query are equally complex (Xue et al., 2024; Liu et al., 2025; Yang et al., 2025). Many intermediate steps in a chain-of-thought process are simple enough for an SLM to handle correctly; however, critical steps—particularly those involving complex calculations or deeper logical inference—pose significant challenges and are more error-prone for smaller models. If we can precisely identify these critical points and selectively engage the stronger reasoning capabilities of an LLM to guide or replace uncertain steps, the remaining reasoning steps performed by the SLM can be corrected, ultimately leading to accurate final predictions.

To address this limitation, we propose selective scaffolding during the reasoning process rather than post-hoc correction of a completed trajectory. As the SLM generates reasoning steps sequentially, each step is assessed for reliablity. If a step $r_i$ is identified as unreliable, an LLM guides immediately, replacing it with a revised step $r_i'$: $r_i' \sim P_{\text{LLM}}(r_i' \mid Q, r_{<i})$. This assistance directly influences all subsequent steps, leading to a new reasoning trajectory $R'$ that builds upon the corrected information.

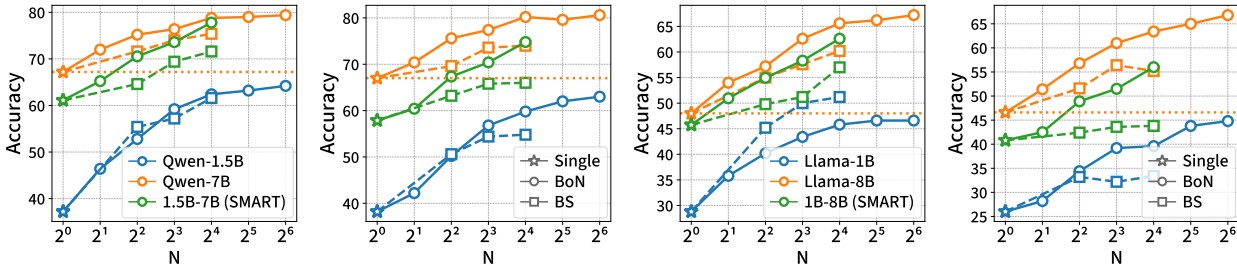

(a) Qwen family with PRM. (b) Qwen family with TLC. (c) Llama family with PRM. (d) Llama family with TLC.

Figure 2: Performance results under varying test-time scales ($N$) using different search strategies. Circles and squares represent results from Best-of-N and Beam Search, respectively. The orange dotted horizontal line indicates the pass@1 performance of the LLM. Average values of 16 samples are reported.

## 2.3 Method

Our framework, termed as $\underline{Sm}$all $\underline{R}$easons, Large Hin$\underline{t}$s (SMART), refines the reasoning process of an SLM by selectively intervening at critical decision points. The method follows:

**1. SLM-generated reasoning draft** The SLM generates an initial reasoning trajectory $R = (r_1, r_2, \ldots, r_m)$ autoregressively, conditioned on the query $Q$: $R \sim P_{\mathrm{SLM}}(R \mid Q)$. Since SLMs lack robust reasoning capabilities, some steps in $R$ may be incorrect or uncertain, leading to errors that propagate throughout the trajectory.

**2. Score-based step evaluation** To identify unreliable reasoning steps, we introduce a step-wise scoring function that assigns a score $s(r_i | r_{<i}, Q)$ to each step $r_i$. We consider two scoring methods:

A. PRM score: A learned reward model evaluates the correctness of each step based on prior context, assigning a score $s(r_i | r_{<i}, Q) \in [0, 1]$.

B. Token-level confidence (TLC): Instead of an external reward model, the token-level confidence of $r_i$ is estimated with its averaged token probability: $s(r_i | r_{<i}, Q) = \frac{1}{n} \sum_{j=1}^{n} P(x_j \mid x_1, \ldots, x_{j-1}, r_{<i}, Q)$ for each token $x_i$ in $r_i$.

During the autoregressive generation, each step is assessed in real-time for score evaluation. Steps with low scores indicate erroneous reasoning or uncertainty, and are candidates for correction.

**3. LLM-based step correction** If step $r_i$'s score falls below a threshold $\tau$, we replace it with an LLM-generated alternative: $r_i' \sim P_{\mathrm{LLM}}(r_i' \mid Q, r_{<i})$. This selective correction improves reasoning quality with only necessary interventions. See Appendix A for algorithmic details.

**Scalability** We briefly demonstrate how it can be integrated with test-time compute scaling methods.

- **Best-of-N**: Multiple reasoning trajectories are generated in parallel. SMART is applied to each trajectory independently, ensuring that erroneous steps are corrected within each sampled path.

- **Beam Search**: Given $N$ candidate sequences and a beam width of $M$, at each reasoning step, the top $M$ sequences are retained based on their scores. Each step $r_i$ within a top sequence is evaluated, and if any candidate node falls below the predefined confidence threshold $\tau$, it is replaced with an LLM-generated alternative.

## 3 Experiments

### 3.1 Experimental setup

`SMART` is evaluated on the MATH500 dataset[1]. (SLM, LLM) pairs for architectures are tested with (Qwen2.5-1.5B, Qwen2.5-7B) and (Llama3.2-1B, Llama3.2-8B). For both pairs, we employ a process reward model with RLHFlow/Llama3.1-8B-PRM-Deepseek-Data (Dong et al., 2024). Stochastic decoding is implemented with temperature equal to 0.8 following Snell et al. (2024). We report Weighted@N accuracy. Threshold values $\tau$ for PRM score and TLC are mainly set to 0.9 and 0.93. More details regarding the design choices are deferred to Appendix B.

### 3.2 Main results

**`SMART` achieves LLM's pass@1 performance more quickly than SLM.** Figure 2 shows the performance of `SMART` across different models and scoring methods as the number of completions $N$ increases under Best-of-N and Beam Search. The orange dotted horizontal line indicates the LLM's pass@1 performance. Unlike approaches that simply scale up the number of sampled reasoning trajectories, often incurring heavy computational costs and relying on unguided exploration, `SMART` focuses on selectively identifying and correcting erroneous reasoning steps within trajectories. This targeted intervention enables `SMART` to rapidly approach or even surpass the LLM-level accuracy at small $N$ values, notably outperforming the LLM accuracy at $N = 2$ or $N = 4$ in most cases. Moreover, Best-of-N consistently outperforms Beam Search across all settings, likely due to more flexible exploration and independent correction of candidate reasoning paths. In contrast, SLM rarely reaches LLM performance, doing so only once at $N = 8$. These results highlight that selectively refining critical points in the reasoning process achieves better accuracy more efficiently than brute-force exploration, making `SMART` a practical and scalable framework for high-quality reasoning. We also show effectiveness of our approach with different models (Gemma2) and domains (Chemistry, Computer Science) in Section C.3.

**`SMART` approaches LLM-level accuracy as test-time compute increases.** Figure 2 also presents that the benefits of `SMART` become increasingly evident as test-time compute scales. With more completions under Best-of-N or Beam Search, `SMART` consistently approaches LLM-level accuracy, demonstrating that additional compute narrows the performance gap. This highlights how selective scaffolding efficiently guides reasoning, reducing the need for exhaustive computation or full trajectory evaluation. Although larger $N$ yields better results, marginal gains diminish at higher values, indicating that `SMART` achieves strong improvements with moderate compute without relying on exhaustive search. Both PRM- and TLC-based scoring methods follow similar scaling trends, with PRM outperforming at higher $N$.

**`SMART` is particularly effective in solving difficult problems.** Table 1 demonstrates that `SMART` achieves relative accuracies exceeding 90% of full LLM performance at higher difficulty levels (Lv 4 and Lv 5), showing very close alignment even on the hardest problems. The improvements over the baseline SLM, up to +140%, highlights substantial gains, with larger gains observed as problem difficulty increases. For instance, using Qwen model with Best-of-N, `SMART` improves accuracy over the SLM by 5.2% at Lv 1 and by 73.2% at Lv 5. This confirms that `SMART`'s selective use of LLM guidance yields the greatest benefits where reasoning is most challenging. Exact accuracy values are shown in Section C.1. Other results with different $N$ or different model, Llama, are depicted in Section C.2.

### 3.3 Analysis of LLM intervention in `SMART`

We analyze the impact of LLM scaffolding by quantifying the extent of intervention in the reasoning process. In doing so, we aim at verifying whether `SMART` intervenes only when and where it is most needed. To this end, we introduce two key metrics—the corrected step ratio and the corrected token ratio—which measure the frequency and extent of LLM modifications, respectively.

---

[1]MATH500 contains a subset of 500 problems from the MATH benchmark(Lightman et al., 2023a) and is provided by Huggingface(https://huggingface.co/datasets/HuggingFaceH4/MATH-500).

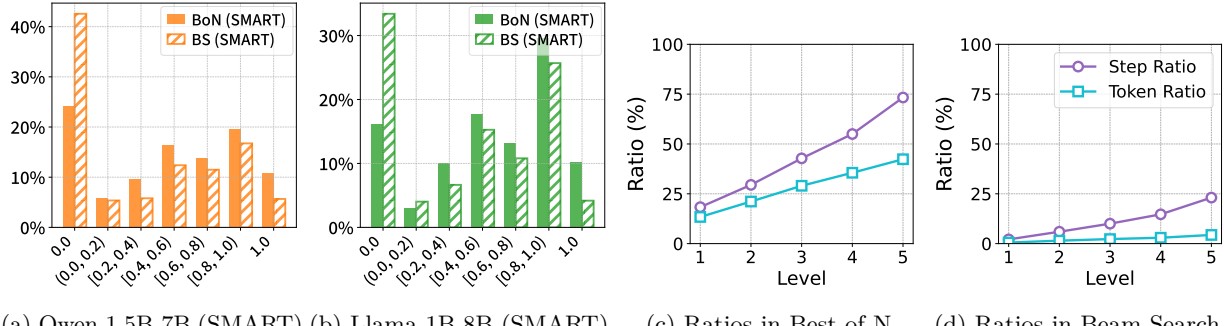

(a) Qwen 1.5B-7B (SMART) (b) Llama 1B-8B (SMART)   (c) Ratios in Best-of-N   (d) Ratios in Beam Search

Figure 3: (Left) Step ratio distribution across (a) Qwen and (b) Llama models upon different search strategies. We report average values of 16 samples. (Right) Comparison of corrected step and token ratios using (c) Best-of-N and (d) Beam Search strategies according to problem difficulty. Average values of 16 samples are reported with Qwen family.

Let $\mathcal{S} = \{1, \cdots, m\}$ be the set of all reasoning steps generated by the SLM. We define the corrected steps as a subset $\mathcal{S}_c \subseteq \mathcal{S}$, where each corrected step $r'_i$ for $i \in \mathcal{S}_c$ is a modification of the original step $r_i$ due to LLM scaffolding. The *corrected step ratio* is then defined as:

$$\text{(Corrected step ratio)} := \frac{|\mathcal{S}_c|}{|\mathcal{S}|}, \tag{3}$$

which reflects how frequently the LLM intervenes in the reasoning trajectory. For Beam Search, we only consider the final reasoning steps, the pruned steps is not included in *corrected step ratio*.

To measure *extent* of these corrections, we also introduce the *corrected token ratio*. Given a tokenization function $\text{Token}(\cdot)$ that returns the token count of a reasoning step, we compute the corrected token ratio as:

$$\text{(Corrected token ratio)} := \frac{\sum_{l \in \mathcal{S}_c} \text{Token}(r'_l)}{\sum_{j \in \mathcal{S}} \text{Token}(r_j)}. \tag{4}$$

This metric captures how extensively the LLM's interventions rewrite the original reasoning content.

**The LLM does not intervene most of the times, engaging dynamically only as needed.** Figure 3a and Figure 3b shows the distribution of corrected step ratios for each model family and search method. Most samples have zero LLM intervention, indicating `SMART` allows the SLM to work independently. Beam Search results concentrate at zero correction, indicating that its early pruning of less-promising candidates effectively limits error propagation within constrained search space, thereby reducing the need for extensive LLM corrections. In contrast, Best-of-N has a broader distribution with many samples showing high correction ratios, suggesting more frequent LLM interventions.

**The LLM intervenes more as the problem becomes more difficult.** Figure 3c and Figure 3d shows the corrected step ratios and token ratios across different problem difficulty levels. At lower difficulty levels, minimal LLM intervention is needed, reflected in low correction ratios. However, as difficulty increases (Lv 4 and 5), LLM intervention becomes more frequent, indicating effective detection and selective guidance when the SLM struggles. Notably, LLM intervention, which is measured by both step and token ratios, is significantly lower under Beam Search compared to Best-of-N, even at higher difficulty levels. This difference arises because Beam Search's tree structure allows a single correction at a parent node to propagate to multiple child paths, while Best-of-N treats each path independently, requiring separate interventions. This pattern holds consistently across model families.

**LLM typically intervenes at the first step of structured reasoning.** Beyond simply adjusting how often scaffolding occurs, determining at which step in the reasoning process to intervene may also be

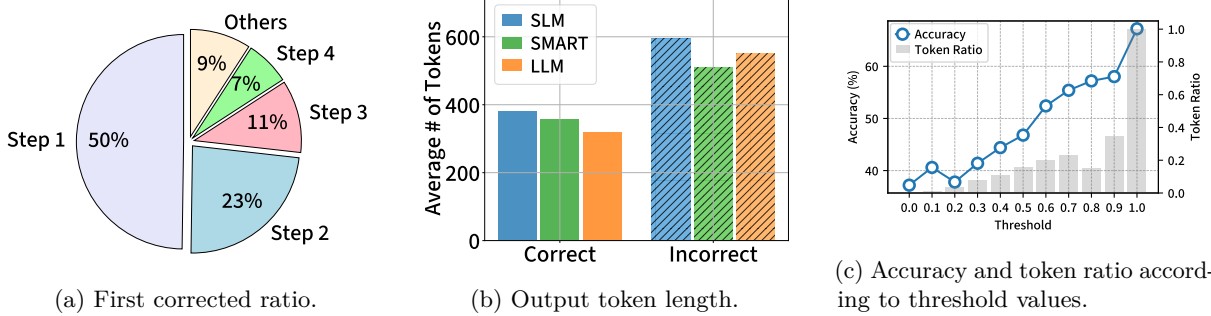

(a) First corrected ratio.

(b) Output token length.

(c) Accuracy and token ratio according to threshold values.

Figure 4: (a) The ratio of SLM's reasoning steps first corrected by the LLM during reasoning process. We use 16 single-generation samples using Qwen models. (b) Average number of tokens by models and correctness using the Best-of-N approach with Qwen models. (c) The relationship between accuracy and token ratio according to threshold values. Qwen model and Best-of-N with PRM score is used.

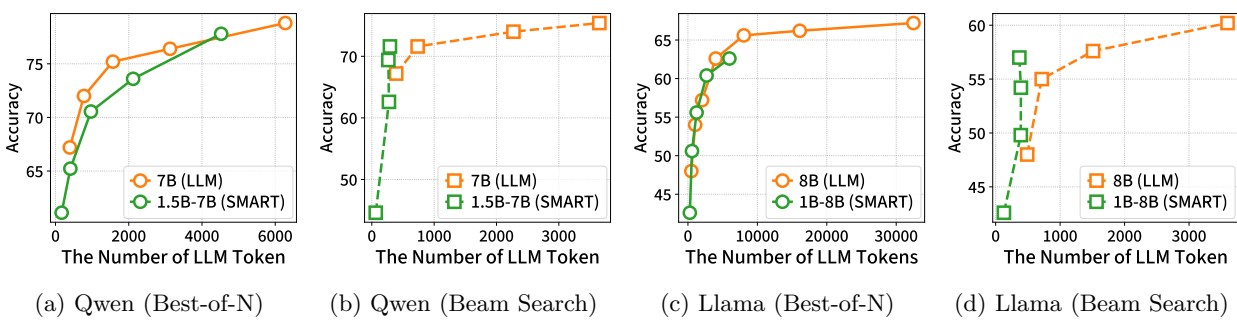

(a) Qwen (Best-of-N)      (b) Qwen (Beam Search)      (c) Llama (Best-of-N)      (d) Llama (Beam Search)

Figure 5: Accuracy according to the number of LLM tokens used during inference. Circles and squares represent results from Best-of-N and Beam Search, respectively. We use PRM score for the experiments.

crucial. In our implementation, the scaffolding is triggered by a score function that does not account for time-step dependencies, but real-world scenarios may well exhibit such dependencies. Indeed, our analysis in Figure 4a shows that the initial steps in the reasoning sequence receives the greatest amount of LLM guidance, suggesting that the LLM effectively identifies and rectifies errors in the early stages of reasoning. This also well aligns with the intuition that early assistance can be particularly beneficial, as unresolved errors may propagate through subsequent steps to degrade reasoning quality. Qualitative examples using `SMART` are presented in Section D.3.

**`SMART` reduces output token length for wrong answers.** Figure 4b illustrates the average number of tokens generated by three different models (SLM, `SMART`, and LLM), distinguishing between correct and incorrect responses. The x-axis represents correctness, while the y-axis indicates the average number of tokens used. Overall, we observe a trend where incorrect responses (hatched bars) tend to have a higher token count across all models. This suggests that models generally generate longer responses when their predictions are incorrect, possibly due to increased uncertainty or excessive generation in ambiguous scenarios. Interestingly, several concurrent works have also reported that reasoning trajectories tend to be longer when models arrive at incorrect answers, reflecting hesitation or over-exploration (Chen et al., 2024b; Marjanović et al., 2025). In contrast, `SMART` exhibits a more concise generation pattern even in failure cases, suggesting a form of disciplined reasoning with reduced verbosity when uncertain.

### 3.4 Efficiency result

In the previous sections, we demonstrated that `SMART` effectively bridges the performance gap between an SLM and an LLM by selectively integrating LLM-generated reasoning steps in a controlled manner. Our results indicate that SLMs, despite their limited reasoning capacity, can achieve near-LLM performance

with minimal but well-targeted LLM scaffolding. This suggests a viable deployment strategy where an SLM operates as the primary model, with an LLM providing corrective reasoning only when necessary.

**SMART reduces LLM token usage significantly while preserving LLM performance.** As illustrated in Figure 5, SMART effectively reduces LLM token usage while maintaining comparable accuracy to an LLM alone. For the Best-of-N strategy, our approach shows efficiency similar to directly employing an LLM. However, with Beam Search, we consistently observe substantial reductions in LLM token usage—*up to 90%*—without sacrificing accuracy. This improvement occurs because the LLM intervenes selectively, only when the SLM's top-ranking candidate reasoning paths fail to meet a specified confidence threshold, thus minimizing overall LLM intervention. Furthermore, as N increases, the number of LLM tokens utilized by SMART remains constant in low number, due to the fixed number (M) of top-ranking paths. These observations suggest that our framework is particularly favorable for tree-structured search strategies, highlighting its potential to significantly reduce token-based costs associated with LLM API usage.

## 4 Discussion

### 4.1 Practical scenario of SMART

**SMART is a practical solution for collaboration between on-device and cloud systems.** A key application of this framework is in scenarios where an SLM runs locally on a device, while an LLM is accessible remotely via an API. Recently, such collaborative frameworks have drawn increased attention due to their potential to simultaneously optimize both accuracy and resource efficiency Narayan et al. (2025). However, due to constraints such as cost, latency, or privacy, frequent LLM queries may be impractical, making it crucial to minimize intervention while maintaining strong performance. SMART addresses this challenge by enabling an adaptive mechanism that dynamically determines when an LLM query is needed, thereby reducing unnecessary API calls while preserving accuracy.

### 4.2 PRM vs. TLC scores

As shown in Figure 2, PRM outperforms TLC at higher values of $N$, primarily because PRM explicitly evaluates the correctness of each reasoning step using predefined criteria, making it more interpretable and controllable. However, PRM is more challenging to apply across diverse domains (Zeng et al., 2025). In contrast, TLC relies on the model's internal confidence, offering greater robustness and generalization, though it cannot capture error propagation in CoT reasoning—where a highly confident but incorrect step may mislead subsequent steps. Despite these limitations, self-confidence scoring methods like TLC remain an active area of research with promising advances in model self-awareness. When computational resources permit, PRM remains the preferred method due to its structured and interpretable evaluation.

### 4.3 Threshold selection

Further optimization of LLM scaffolding strategies remains an open challenge. Currently, the scaffolding rate is controlled by a fixed threshold that determines when an SLM-generated step should be replaced. As shown in Figure 4c, there is an inherent trade-off between accuracy and the proportion of reasoning tokens contributed by the LLM. A promising direction for future research is to develop reinforcement learning or meta-learning approaches to dynamically adjust scaffolding thresholds based on problem complexity and past performance.

## 5 Related works

**Application of SLMs** SLMs are increasingly employed in resource-constrained environments, such as mobile devices and embedded systems, where deploying LLMs is impractical due to computational and memory limitations Blog (2024). SLMs' efficiency makes them well-suited for real-time applications, including chatbots, live captioning, and gaming interactions Xu et al. (2025). They are also utilized in cloud-integrated systems, where they enhance automation and personalized services while reducing infrastructure costs Talluri

et al. (2024). Despite these advantages, SLMs face significant limitations, particularly in structured reasoning tasks Bi et al. (2024). Their constrained capacity leads to performance saturation during training, resulting in struggles with nuanced language comprehension and reduced accuracy on complex tasks (Godey et al., 2024; Yi et al., 2024).

**Multi-LM collaboration**  Collaborative decoding enables multiple language models to work together to enhance text generation (Shen et al., 2024a). At the token level, different models contribute at various points in the sequence by employing SLM to draft tokens and LLM to verify them for efficient decoding (Leviathan et al., 2023), utilizing different expert models aligned with specific tasks to improve performance (Liu et al., 2021), or integrating a verifier model to refine generation (Lightman et al., 2023b). To address complex reasoning tasks, collaborative decoding has evolved to the step level. For hallucination mitigation, a primary model generates reasoning steps while another model verifies their validity (Feng et al., 2024). In multi-agent setups, multiple smaller models assume different roles, contributing individual insights in a structured debate. Arguments are then synthesized to reach a final conclusion (Shen et al., 2024b). While the overall concept of `SMART` might be similar with speculative decoding in terms of utilizing SLM and LLM in collaboration, the objective and purpose is fundamentally different, and we elaborate this in Section D.2.

**Test-time compute scaling**  Recent work has explored test-time compute scaling as an alternative to fine-tuning or distillation, enhancing SLM reasoning by generating multiple reasoning paths or refining outputs iteratively (Snell et al., 2024; Ehrlich et al., 2025; Muennighoff et al., 2025). While increasing test-time computation improves performance, SLMs inherently lack the capacity to solve complex reasoning problems, even with extensive compute scaling. This fundamental limitation stems from their restricted expressivity, making it impossible for SLMs to reach LLM-level reasoning solely through brute-force search or increased sampling. Additionally, test-time scaling incurs higher computational costs, limiting its practicality for real-time applications.

**Confidence-based score**  Recent studies have demonstrated that the intrinsic confidence of a model's predictions can serve as a reliable indicator of reasoning quality. For instance, Wang & Zhou (2024) found that in large language models, the presence of a chain-of-thought reasoning path correlates with higher confidence in the model's decoded answer. Moreover, Huang et al. (2025) introduced a self-calibration method that dynamically adjusts the number of sampling responses based on the model's confidence in its predictions.

# 6    Conclusion

This paper presents a novel `SMART` (Small Reasons, Large Hints) framework that enhances structured, multi-step reasoning by selectively incorporating LLM interventions into an SLM's reasoning process. `SMART` dynamically assesses each reasoning step using a score-based mechanism and replaces unreliable steps with LLM-generated alternatives only at critical junctures. Experimental results indicate that `SMART` can achieve up to 98% of LLM performance while significantly reducing LLM token usage up to 90%. This suggests a viable deployment strategy where an SLM functions as the primary model, with an LLM providing corrections only when necessary—a practical avenue for deploying SLMs where on-device operation is desired—thereby avoiding the overhead of exhaustive compute scaling.

**Broader Impact Statement**

Our work, while designed to enhance the reasoning capabilities of smaller models, may introduce a potential risk of bias propagation. Specifically, any inherent biases and errors from the LLM can be encoded into the generated reasoning plans and subsequently adopted by the student model. Users of this research should therefore be aware that the final outputs may reflect these inherited biases, and careful consideration is needed to audit the reasoning pathways provided by the guiding LLM.

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

# A  Implementation details of `SMART`

---

**Algorithm 1** `SMART`

---

**Require:** Query $Q$, max steps $M$, scoring function $s(\cdot)$, threshold $\tau$, Token count function Token$(\cdot)$, max token length $L_{\max}$

**Ensure:** Reasoning sequence $R = (r_1, \ldots, r_m)$ and final answer $A = \phi(Q, R)$

1: $R \leftarrow []$
2: **for** $i = 1$ to $M$ **do**
3:     $r_i^{(SLM)} \leftarrow P_{\mathrm{SLM}}(r_i \mid Q, r_1, \ldots, r_{i-1})$
4:     $s \leftarrow s(r_i^{(SLM)} | Q, R)$
5:     $r_i \leftarrow \begin{cases} P_{\mathrm{LLM}}(r_i \mid Q, r_1, \ldots, r_{i-1}) & \text{if } s < \tau, \\ r_i^{(SLM)} & \text{otherwise.} \end{cases}$
6:     Append $r_i$ to $R$
7:     **if** $r_i = $ EOS or $\sum_{j=1}^{i}$ Token$(r_j) \geq L_{\max}$ **then**
8:         **break**
9:     **end if**
10: **end for**
11: **return** $R$

---

We present the implementation details of our `SMART` framework, which selectively integrates LLM intervention to enhance the structured reasoning capabilities of SLMs. The reasoning process is formulated as an iterative decision-making problem, where the SLM generates reasoning steps autonomously, and an LLM intervenes selectively based on score-based evaluation.

## A.1  Reasoning Process

Given an input query $Q$, the objective is to construct a reasoning trajectory $R = (r_1, \ldots, r_m)$ that leads to a final answer $A$. The `SMART` framework operates as follows:

1. **SLM Step Generation:** The SLM generates an intermediate reasoning step $r_i^{(SLM)}$ autoregressively:

$$r_i^{(SLM)} \sim P_{\mathrm{SLM}}(r_i \mid Q, r_1, \ldots, r_{i-1}) \tag{5}$$

    where $P_{\mathrm{SLM}}$ represents the probability distribution modeled by the SLM.

2. **Step Scoring:** Each generated step is evaluated with a scoring function $s(r_i^{(S)}|Q, r_{<i})$, which assesses its reliability based on predefined heuristics or reward models.

3. **Selective LLM Intervention:** If the score $s$ is lower than the threshold $\tau$, the reasoning step is corrected by querying the LLM:

$$r_i' \sim P_{\mathrm{LLM}}(r_i \mid Q, r_1, \ldots, r_{i-1}) \tag{6}$$

$$r_i \leftarrow r_i' \tag{7}$$

    Otherwise, the SLM-generated step is retained.

    After that, append reasoning step $r_i$ to reasoning trajectory $R$

4. **Termination Criteria:** The iterative reasoning process continues until one of the following conditions is met:

   - The model generates an **end-of-the-sequence (EOS)** token.
   - The cumulative token count surpasses $L_{\max}$, ensuring computational efficiency.

5. **Output Reasoning Trajectory:** The final reasoning sequence $R$ is returned as the output.

# B    Experimental details

## B.1    Detailed experiment setup

We use Qwen family (Qwen2.5-1.5B, Qwen2.5-7B) and Llama family (Llama3.2-1B, Llama3.2-8B) in our experiments. For experiments, we use four A5000 GPUs using inference package VLLM (Kwon et al., 2023) 0.6.3. We build our implementation on top of the base code provided by Face (2024) with Apache-2.0 license.

## B.2    Dataset details

The MATH500 dataset is a curated subset of the MATH dataset, comprising 500 problems selected from MATH dataset(Lightman et al., 2023a), to evaluate mathematical reasoning in models. These problems are categorized by subject and difficulty level, facilitating a comprehensive assessment of model performance across various mathematical domains. Each problem is accompanied by a detailed step-by-step solution, enabling the evaluation of models' problem-solving processes. In our experiments, we utilize the MATH500 dataset to benchmark the performance of our models, ensuring a rigorous evaluation of their mathematical reasoning abilities.

## B.3    Threshold selection for `SMART`

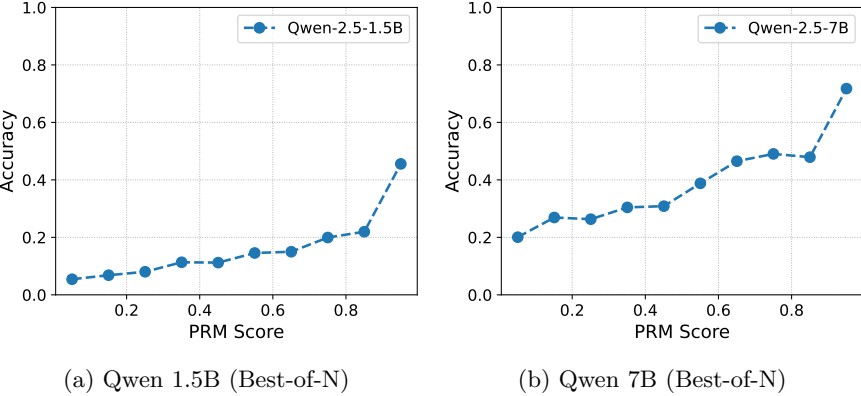

(a) Qwen 1.5B (Best-of-N)                 (b) Qwen 7B (Best-of-N)

Figure 6: Accuracy according to PRM score. Qwen-2.5-1.5B and Qwen-2.5-7B are used for predicting reasoning steps.

In Figure 6, the relationship between the PRM score and prediction accuracy differs notably between the two model sizes. For the 7B model (Qwen-2.5-7B) in Figure 6b, the PRM score strongly correlates with accuracy, indicating that higher PRM values reliably predict correct outcomes. In contrast, for the 1.5B model (Qwen-2.5-1.5B) shown in Figure 6a, accuracy increases only marginally across most PRM score ranges and remains low until approximately 0.9. Based on this observation, we set a PRM threshold of 0.9 to trigger LLM intervention, as reasoning steps with a score of 0.9 or lower exhibit a high likelihood of being incorrect, necessitating the intervention of the LLM.

## B.4    Evaluation metrics

The Weighted@N metric selects the answer with the highest total reward by aggregating scores across identical responses. It prioritizes high-quality outputs by reinforcing frequently occurring, high-reward solutions. Formally, the selected answer $A$ is given by:

$$A_{\text{weighted}} = \arg\max_a \sum_{i=1}^{N} \mathbb{I}(A_i = a) \cdot s(R_i \mid Q)$$

where $s(R_i \mid Q)$ is a scoring function which evaluates the correctness of reasoning trajectory $R_i$ given query $Q$. The trajectory $R_i$ consists of a sequence of intermediate steps leading to the final answer $A_i$. We report Weighted@N as it shows consistently high performance among other metrics such as majority voting.

### B.5 Zero-shot prompt for evaluation

We use zero-shot prompting method to evaluate following . This prompt instructs to follow a structured step-by-step format, where each step is separated by double line breaks (`\n\n`). This separation facilitates evaluation and correction. The response always concludes with a boxed final answer for clarity.

---

**Zero-shot Prompt**

<|im_\$start|>system
Solve the following math problem efficiently and clearly:
- For simple problems (2 steps or fewer): Provide a concise solution with minimal explanation.
- For complex problems (3 steps or more): Use this step-by-step format:
`## Step 1:` **[Concise description]** [Brief explanation and calculations]
`## Step 2:` **[Concise description]** [Brief explanation and calculations]
...
Regardless of the approach, always conclude with:
Therefore, the final answer is: $\boxed{answer}$. I hope it is correct.
Where [answer] is just the final number or expression that solves the problem. <|im_\$end|>
<|im_\$start|>user
{Problem} <|im_\$end|>
<|im_\$start|>assistant

---

## C   Additional results

### C.1   Performance result of `SMART`

We show extended results for Table 1. Table 2 reports the performance values using `SMART` on different models, search strategies, and test-time scales ($N$). As in , `SMART` reaches near-LLM accuracy, and the effectiveness increases for more complex problems.

### C.2   Comparison of `SMART` with the target LLM

We further evaluate `SMART`'s performance on Qwen family (Table 3) and extend the analysis to the Llama model family (Table 4). The overall trend remains consistent—`SMART` achieves near-LLM accuracy at lower difficulty levels, exceeding 90% even for moderate N. As task difficulty increases, performance remains competitive. While Beam Search provides structured decoding, Best-of-N tends to yield higher gains, particularly at higher difficulty levels. Notably, for Llama, `SMART` occasionally surpasses LLM performance, highlighting its robustness across models and search strategies.

### C.3   Generalizability of `SMART` on other domains and models

Most of the test-time reasoning study focuses only on the mathematical or coding domain Snell et al. (2024); Muennighoff et al. (2025) since it has a clear ground truth and standard to assess its reasoning steps. However, generalizing our findings beyond mathematical reasoning can be an interesting direction. Furthermore, checking our framework's generalizability to diverse architecture is needed. We have conducted additional experiments across a different reasoning domains and with diverse model architectures.

**Difference domain** In Table 5 (Left), we evaluated our framework on the MMLU-Pro dataset Wang et al. (2024), a benchmark designed for assessing the multi-task language understanding and reasoning capabilities of models across a diverse array of academic subjects. We experiment on Chemistry and Computer science

Table 2: Performance of SMART method using different models, search strategies, and *N* according to the difficulty level of problem on the MATH500 dataset. The values in parentheses indicate the improvement over the SLM baseline, with colors denoting the direction of change. Performance values exceeding 90.00 are underlined, and the improvement larger than 10.00 are bold-faced. Average values of 16 samples are reported.

| Model | Search | N | Lv 1 | Lv 2 | Lv 3 | Lv 4 | Lv 5 |
|---|---|---|---|---|---|---|---|
| Qwen2.5 7B / Qwen2.5 1.5B | Single | 1 | 79.79 (+12.94) | 76.39 (+17.60) | 72.49 (+24.33) | 57.43 (+23.80) | 36.84 (+21.34) |
| | Best-of-N | 2 | 83.41 (+8.13) | 80.89 (+13.74) | 76.00 (+19.94) | 62.16 (+21.34) | 41.09 (+22.48) |
| | | 4 | 87.50 (+4.68) | 85.01 (+9.31) | 83.20 (+18.56) | 68.08 (+18.86) | 45.61 (+22.48) |
| | | 8 | 89.50 (+3.45) | 87.23 (+4.48) | 87.40 (+15.25) | 73.05 (+16.20) | 47.98 (+22.03) |
| | | 16 | 91.85 (+3.45) | 90.00 (+6.10) | 89.50 (+11.90) | 76.95 (+19.55) | 50.35 (+20.50) |
| | | 32 | 93.00 (+4.60) | 91.10 (+5.50) | 88.60 (+7.60) | 79.70 (+21.10) | 51.50 (+20.90) |
| | Beam Search | 4 | 83.70 (-4.70) | 80.00 (+3.30) | 78.10 (+14.30) | 60.90 (+5.40) | 40.30 (+16.40) |
| | | 8 | 86.00 (-2.40) | 81.10 (+0.00) | 78.10 (+5.70) | 64.10 (+13.30) | 44.80 (+19.40) |
| | | 16 | 93.00 (+2.30) | 85.60 (+2.30) | 81.90 (+5.70) | 64.80 (+9.30) | 47.00 (+14.90) |
| Llama3.1 8B / Llama3.2 1B | Single | 1 | 79.58 (+20.72) | 66.66 (+20.10) | 55.40 (+21.15) | 40.56 (+20.63) | 20.22 (+11.52) |
| | Best-of-N | 2 | 84.59 (+10.87) | 72.56 (+16.52) | 61.84 (+20.09) | 45.96 (+21.24) | 24.21 (+13.45) |
| | | 4 | 85.74 (+13.41) | 78.34 (+17.43) | 67.86 (+20.23) | 50.10 (+19.82) | 28.00 (+18.10) |
| | | 8 | 88.35 (+11.65) | 82.78 (+19.00) | 72.85 (+19.55) | 54.68 (+19.98) | 30.05 (+20.95) |
| | | 16 | 89.55 (+12.25) | 86.70 (+16.13) | 76.65 (+20.75) | 58.20 (+21.60) | 34.70 (+20.40) |
| | | 32 | 90.70 (+15.40) | 85.60 (+15.10) | 81.90 (+26.40) | 59.40 (+25.50) | 36.60 (+25.10) |
| | Beam Search | 4 | 83.00 (-4.00) | 70.00 (+7.80) | 61.90 (+11.40) | 43.00 (+6.30) | 22.40 (+6.00) |
| | | 8 | 81.40 (-7.00) | 74.40 (+4.40) | 59.00 (-3.90) | 43.00 (+3.20) | 26.90 (+10.50) |
| | | 16 | 83.70 (+0.00) | 81.10 (+10.00) | 69.50 (+5.70) | 50.00 (+9.00) | 29.10 (+8.20) |

Table 3: Comparison of SMART with the target LLM (used as the teacher in scaffolding) for the Qwen model family, using different search methods. Each cell shows the percentage of our performance relative to the target LLM (i.e., 100% means SMART brings the identical performance with that of LLM). Values in parentheses indicate the relative performance of SLM against the target LLM. Performance values over 90% are underlined.

| Approach | N | Lv 1 | Lv 2 | Lv 3 | Lv 4 | Lv 5 |
|---|---|---|---|---|---|---|
| Best of N | 1 | 89.93% (75.35%) | 89.00% (68.49%) | 89.21% (59.27%) | 90.98% (53.28%) | 90.84% (38.22%) |
| | 2 | 91.68% (82.75%) | 91.21% (75.72%) | 89.66% (66.13%) | 91.70% (60.22%) | 95.55% (43.28%) |
| | 4 | 94.67% (89.61%) | 92.90% (82.72%) | 94.20% (73.18%) | 93.43% (67.54%) | 95.50% (48.44%) |
| | 8 | 100.0% (95.56%) | 97.50% (92.99%) | 96.99% (88.43%) | 98.24% (88.56%) | 98.26% (75.69%) |
| | 16 | 100.0% (96.89%) | 98.65% (93.53%) | 98.56% (89.17%) | 98.61% (85.22%) | 99.12% (81.15%) |
| | 32 | 100.0% (99.32%) | 99.21% (96.74%) | 99.19% (94.55%) | 98.43% (90.54%) | 99.00% (85.10%) |
| Beam Search | 4 | 92.28% (97.46%) | 88.89% (85.22%) | 93.20% (76.13%) | 83.77% (76.34%) | 84.31% (50.00%) |
| | 8 | 92.47% (95.05%) | 87.96% (87.96%) | 91.13% (84.48%) | 95.39% (75.60%) | 79.01% (44.80%) |
| | 16 | 100.00% (97.53%) | 93.96% (91.44%) | 92.44% (86.01%) | 87.33% (74.80%) | 92.70% (63.31%) |

with 300 random test samples each. We use Llama model family, VersaPRM for PRM Zeng et al. (2025) and 3-shot prompt.

**Difference model architecture** We employed different model architectures, including Gemma2-2b and Gemma2-9b, to assess whether our method can improve the performance for different model architectures. Table 5 (Right) summarizes the performance comparison of our approach versus baseline methods on the Gemma2 models on MATH500.

Table 4: Comparison of SMART with the target LLM (used as the teacher in scaffolding) for the Llama model family, using different search methods. Each cell shows the percentage of our performance relative to the target LLM (i.e., 100% means SMART brings the identical performance with that of LLM). Values in parentheses indicate the relative performance of SLM against the target LLM. Performance values over 90% are underlined.

| Approach | N | Lv 1 | Lv 2 | Lv 3 | Lv 4 | Lv 5 |
|---|---|---|---|---|---|---|
| Best of N | 1 | 97.43% (72.06%) | 96.52% (67.42%) | 97.66% (60.38%) | 103.06% (50.64%) | 104.91% (45.14%) |
| | 2 | 97.82% (79.16%) | 95.14% (72.49%) | 97.26% (65.72%) | **102.28%** (55.00%) | 104.90% (47.54%) |
| | 4 | 96.39% (81.35%) | 97.59% (77.16%) | 94.07% (66.19%) | 97.14% (58.70%) | 102.75% (52.11%) |
| | 8 | **99.33%** (88.87%) | 100.67% (78.38%) | **102.56%** (66.62%) | **104.21%** (67.09%) | 92.53% (47.73%) |
| | 16 | **102.60%** (89.48%) | 101.42% (88.15%) | 98.86% (71.14%) | 97.12% (70.45%) | 94.94% (45.01%) |
| Beam Search | 4 | 94.68% (100.00%) | 88.72% (78.83%) | 98.41% (80.29%) | 87.40% (74.59%) | 81.16% (59.42%) |
| | 8 | 94.65% (102.79%) | 91.74% (86.31%) | 91.05% (97.07%) | 75.44% (69.83%) | 87.91% (53.60%) |
| | 16 | 92.28% (92.28%) | **97.36%** (85.35%) | 88.99% (81.69%) | 90.09% (73.15%) | **114.57%** (82.28%) |

Table 5: (Left) Performance on MMLU-Pro in two academic domains (Chemistry and Computer Science) using Llama family with 3-shot prompt. SMART shows improved performance over 1B-only baselines. We employ with Beam Search, N=4. (Right) Performance on Gemma2 using MATH500. SMART improves the reasoning performance of the smaller Gemma2-2B model when scaffolded by the larger Gemma2-9B.

| Model | Chemistry | Computer Science |
|---|---|---|
| Llama3.1-8B-Instruct | 14.33 | 16.67 |
| Llama3.2-1B-Instruct | 7.00 | 5.67 |
| Llama3.2-1B-8B (Ours) | 12.33 | 11.67 |

| Model | BoN (N=8) | BS (N=8) |
|---|---|---|
| Gemma2-9B-IT | 54.8 | 55.4 |
| Gemma2-2B-IT | 28.8 | 34.6 |
| Gemma2-2B-9B (Ours) | 48.6 | 50.6 |

# D  Further Discussions

## D.1  The relationship between threshold values of TLC score and accuracy and token ratio

We show extended result of accuracy and token ratio according to TLC score. As illustrated in Figure 7, there exists an clear trade-off between overall accuracy and the proportion of reasoning tokens contributed by the LLM, which has the same observation with Section 4.3.

## D.2  Comparison between SMART and speculative decoding

The common ground of SMART and speculative decoding is that both use SLM and LLM for generation. However, these two frameworks are fundamentally different in following properties:

- **Objective.** Speculative decoding primarily aims at accelerating the decoding speed of a single LLM by using a smaller model as a draft generator to predict future tokens rapidly. Its main focus is latency reduction during generation. SMART, on the other hand, targets improving the structured reasoning quality of SLMs in resource-constrained setting by selectively integrating targeted reasoning corrections from LLMs, thus enhancing reasoning accuracy rather than faster decoding.

- **Usage.** Speculative decoding is generally employed in real-time inference scenarios requiring rapid text generation, such as interactive chatbots or streaming text outputs. SMART is tailored for scenarios involving complex multi-step reasoning, where occasional assistance from a more capable model (LLM) is strategically leveraged to overcome inherent limitations in SLM reasoning. One key application is where an SLM runs locally on a device, while an LLM is accessible remotely via an API.

- **Cost.** In speculative decoding, the smaller model's predictions are continuously validated by the LLM, resulting in consistent and relatively high LLM usage. SMART significantly reduces LLM interaction,

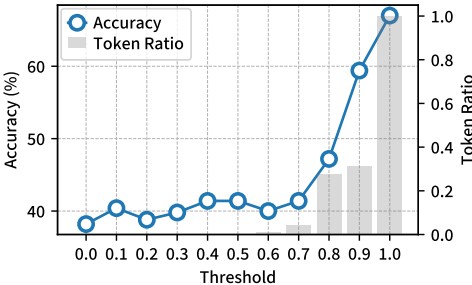

Figure 7: The relationship between accuracy and token usage ratio according to threshold values. Threshold values of TLC score is used.

> invoking it only at critical reasoning points (reducing LLM token usage by up to 90%), thereby providing substantial cost savings in API-based deployments.

Since speculative decoding serves a distinct purpose, speculative decoding can be integrated within the `SMART` framework itself—potentially accelerating the generation steps for both the SLM reasoning trajectories and the LLM's corrective interventions.

### D.3 Qualitative analysis of `SMART`

Table 6 to 10 illustrate where and how the LLM corrects the reasoning steps of the SLM when applying our `SMART` method. The table presents randomly selected samples to demonstrate this correction process.

As previously mentioned, the LLM consistently provides corrections within the initial reasoning steps (i.e., ## Step 1 or 2). Specifically, it identifies logically flawed components in the SLM's reasoning, generates an alternative formulation, and subsequently adjusts the trajectory to ensure that the following reasoning steps align correctly.

## E AI usage

We utilize GPT-4o, O3-mini-high, Copilot, and Perplexity to assist with coding and writing tasks. However, the concepts and structure are formulated by the authors, while AI models are employed primarily for refining code and enhancing the clarity and coherence of the writing.

Table 6: Example of generation using SMART on **Level 1** difficulty problem.

---

```
Question: Suppose $\\sin D = 0.7$ in the diagram below. What is $DE$? [asy
    ]\npair D,E,F;\nF = (0,0);\nD = (sqrt(51),7);\nE = (0,7);\ndraw(D--E--F
    --D);\ndraw(rightanglemark(D,E,F,15));\nlabel("$D$",D,NE);\nlabel("$E
    $",E,NW);\nlabel("$F$",F,SW);\nlabel("$7$",(E+F)/2,W);\n[/asy]

SMART Output: We have two right-angled triangles in the diagram. Let's
    denote the length of $DE$ as $x$ and the length of $DF$ as $h$.\n\n

    (SLM step) From the given $\\sin D = 0.7$, we have:\n\\[ \\frac{h}{7}
        = 0.7 \\]\n\\[ h = 7 \\times 0.7 = 4.9 \\]
    (LLM scaffolding step) ## Step 1: Use the right triangle DFE\nIn $\\
        triangle DFE$, we have $\\sin D = \\frac{\\text{opposite}}{\\text{
        hypotenuse}} = \\frac{7}{h} = 0.7$. So, $h = \\frac{7}{0.7} = 10$.

    ## Step 2: Use the Pythagorean Theorem in triangle DFE\nWe also know
        that in $\\triangle DFE$, $DE^2 + DF^2 = DF^2$. Substituting the
        known values, we get $x^2 + 7^2 = 10^2$. Simplifying, we get $x^2 +
         49 = 100$, so $x^2 = 51$.\n\nTherefore, $x = \\sqrt{51}$.\n\
        nTherefore, the final answer is: $\\boxed{\\sqrt{51}}$.
```

---

Table 7: Example of generation using SMART on **Level 2** difficulty problem.

---

Question: The Greek army contained two types of soldiers: the upper class
    and the lower class soldiers. If there were a total of 5 upper class
    soldiers, and 10 lower class soldiers in a certain part of Athens, and
    the battle of Thermopylae demands a force of 4 upper class soldiers and
    8 lower class soldiers, how many different battalions can be sent?

SMART Output: To solve this problem, let's break it down step by step:\n\n

1. First, we need to determine the total number of soldiers in the given
    part of Athens. We know that there are 5 upper class soldiers and 10
    lower class soldiers. So the total number of soldiers is:\n   \\[5 + 10
    = 15\\]\n\n

    (SLM step) 2. Next, to find out how many different battalions can be
        sent, we need to divide the total number of soldiers by the number
        of soldiers required per battalion. In this case, each battalion
        needs 4 upper class soldiers and 8 lower class soldiers. So we
        divide the total number of soldiers by the sum of these
        requirements:\n   \\[15 \\div (4 + 8) = 15 \\div 12 = 1.25\\]

    (LLM correction step) 2. Next, we need to determine how many different
        battalions can be formed from the upper class soldiers. We need to
        choose 4 upper class soldiers out of 5. This is a combination
        problem, and the number of ways to choose 4 soldiers out of 5 is
        given by the combination formula \\( \\binom{5}{4} \\):\n   \\[\n
        \\binom{5}{4} = \\frac{5!}{4!(5-4)!} = \\frac{5!}{4!1!} = \\frac
        {5 \\cdot 4!}{4! \\cdot 1} = 5\n   \\]

    3. Similarly, we need to determine how many different battalions can
        be formed from the lower class soldiers. We need to choose 8 lower
        class soldiers out of 10. This is also a combination problem, and
        the number of ways to choose 8 soldiers out of 10 is given by the
        combination formula \\( \\binom{10}{8} \\):\n   \\[\n   \\binom
        {10}{8} = \\frac{10!}{8!(10-8)!} = \\frac{10!}{8!2!} = \\frac{10 \\
        cdot 9 \\cdot 8!}{8! \\cdot 2 \\cdot 1} = \\frac{10 \\cdot 9}{2 \\
        cdot 1} = 45\n   \\]\n\n4. Since the battalions are formed
        separately (one from upper class soldiers and one from lower class
        soldiers), the total number of different battalions that can be
        sent is the product of the number of ways to form the battalions
        from each group:\n   \\[\n   5 \\cdot 45 = 225\n   \\]\n\nTherefore
        , the final answer is:\n\\[\n\\boxed{225}\n\\]

---

Table 8: Example of generation using `SMART` on **Level 3** difficulty problem.

```
Question: Evaluate $(1+2i)6-3i$.

SMART Output:
(SLM step) ## Step 1:**\nFirst, we need to distribute the 6 in the
    expression $(1+2i)6-3i$. This means multiplying 6 by each term inside
    the parentheses.
(LLM correction step) ## Step 1: Distribute and simplify the expression\
    nFirst, distribute $6$ across the terms in the parentheses and then
    subtract $3i$.

## Step 2: Perform the multiplication and subtraction\n$6(1 + 2i) - 3i = 6
    + 12i - 3i = 6 + 9i$.\n\nTherefore, the final answer is: $\\boxed{6 +
    9i}$.
```

Table 9: Example of generation using `SMART` on **Level 4** difficulty problem.

Question: What is the probability that a randomly selected integer in the
set $$\\{1,2,3,\\ldots,100\\}$$ is divisible by 2 and not divisible by
3? Express your answer as a common fraction.

SMART Output: To find the probability, we need to consider the following
steps:\n\n

(SLM step) 1. Find the total number of integers in the set.\n2. Find the
number of integers divisible by 2 and not divisible by 3.\n3. Find the
number of integers divisible by 3 and not by 2.\n4. Subtract the result
from step 2 and the result from step 3 from the total to get the
number of integers divisible by 2 and not divisible by 3.\n5. Divide
the result from step 4 by the total number of integers in the set to
get the probability.
(LLM correction step) ## Step 1: Determine the total number of integers in
the set.\nThe set is \\(\\{1, 2, 3, \\ldots, 100\\}\\), so the total
number of integers is \\(100\\).

(SLM step) ## Step 2: Count the number of integers in the set that are
divisible by 2.\nIntegers divisible by 2 in the set are \\(\\{2, 4, 6,
\\ldots, 100\\}\\). This is an arithmetic sequence where the first term
\\(a_1 = 2\\) and the common difference \\(d = 2\\). The nth term of
an arithmetic sequence can be found using the formula \\(a_n = a_1 + (n
-1)d\\). Setting \\(a_n = 100\\) gives the equation \\(100 = 2 + (n-1)
2\\). Solving for \\(n\\) gives \\(n = 50\\). Therefore, there are 50
integers in the set that are divisible by 2.\n\n
(LLM correction step) ## Step 2: Count the numbers divisible by 2.\nA
number is divisible by 2 if it is even. The even numbers in the set are
\\(2, 4, 6, \\ldots, 100\\). This is an arithmetic sequence where the
first term is 2 and the last term is 100, with a common difference of
2. The number of terms in this sequence is \\(\\frac{100}{2} = 50\\).\n
\n

## Step 3: Count the numbers divisible by 2 but not by 3.\nWe need to
exclude the numbers that are divisible by both 2 and 3 (i.e., divisible
by 6) from the count of numbers divisible by 2, because they are
already counted in the previous step. The numbers divisible by 6 in the
set are \\(6, 12, 18, \\ldots, 96\\). This is an arithmetic sequence
where the first term is 6 and the last term is 96, with a common
difference of 6. The number of terms in this sequence is \\(\\frac
{96}{6} = 16\\). Therefore, the number of integers in the set that are
divisible by 2 but not by 3 is \\(50 - 16 = 34\\).\n\n

## Step 4: Calculate the probability.\nThe probability that a randomly
selected integer from the set is divisible by 2 but not by 3 is the
number of such integers divided by the total number of integers in the
set. This gives us \\(\\frac{34}{100} = \\frac{17}{50}\\).\n\nTherefore
, the final answer is: $\\boxed{\\frac{17}{50}}$.'

Table 10: Example of generation using `SMART` on **Level 5** difficulty problem.

---

Question: Bill walks $\\frac{1}{2}$ mile south, then $\\frac{3}{4}$ mile
    east, and finally $\\frac{1}{2}$ mile south. How many miles is he, in a
     direct line, from his starting point?  Express your answer as a
    decimal to the nearest hundredth.

SMART Output: To solve this problem, let's break down Bill's journey into
    a series of steps and then calculate the final distance from his
    starting point.\n\n

(SLM step) 1. **First Leg**: Bill walks $\\frac{1}{2}$ mile south. His
    position is now $-\\frac{1}{2}$ mile South from his starting point.\n2.
     **Second Leg**: Bill then walks $\\frac{3}{4}$ mile east. After this
    leg, he is now $\\frac{3}{4}$ mile North and $\\frac{1}{2}$ mile South
    of his starting point. His net position change in the North-South
    direction is $-\\frac{1}{2} + \\frac{3}{4} = \\frac{1}{4}$ mile North.\
    n3. **Third Leg**: Finally, Bill walks $\\frac{1}{2}$ mile south. After
     this leg, he is now $\\frac{1}{4}$ mile North and $\\frac{1}{2}$ mile
    South of his starting point. His net position change in the South
    direction is $\\frac{1}{2} - \\frac{1}{2} = 0$ mile.
(LLM correction step) ## Step 1: Determine the total southward
    displacement\nBill walks $\\frac{1}{2}$ mile south, then another $\\
    frac{1}{2}$ mile south. So, the total southward displacement is:\n\\[\n
    \\frac{1}{2} + \\frac{1}{2} = 1 \\text{ mile}\n\\]
\n\n## Step 2: Determine the eastward displacement\nBill walks $\\frac
    {3}{4}$ mile east. This represents the eastward displacement.\n\n

(SLM step) ## Step 3: Calculate the resultant displacement\nTo find the
    resultant displacement, we can represent Bill's movements as vectors.
    We have:\n- A southward displacement of $\\frac{1}{2}$ mile\n- A
    southward displacement of $\\frac{1}{2}$ mile (next south)\n- An
    eastward displacement of $\\frac{3}{4}$ mile\n\n

(LLM correction step) ## Step 3: Use the Pythagorean theorem to find the
    direct distance\nNow we can use the Pythagorean theorem to find the
    direct distance from the starting point. The total southward
    displacement and the eastward displacement form the legs of a right
    triangle, with the direct distance being the hypotenuse. The
    Pythagorean theorem states:\n\\[\nc = \\sqrt{a^2 + b^2}\n\\]\nwhere $a
    = 1$ mile (southward displacement) and $b = \\frac{3}{4}$ mile (
    eastward displacement). Plugging in these values, we get:\n\\[\nc = \\
    sqrt{1^2 + \\left(\\frac{3}{4}\\right)^2} = \\sqrt{1 + \\frac{9}{16}} =
     \\sqrt{\\frac{16}{16} + \\frac{9}{16}} = \\sqrt{\\frac{25}{16}} = \\
    frac{5}{4} = 1.25\n\\]\n

Therefore, the final answer is: $\\boxed{1.25}$ miles.

---

