# OpenReview forum: "Guiding Reasoning in Small Language Models with LLM Assistance"
_TMLR — Rejected by TMLR_

### Review · Reviewer_ptUN · 2025-09-22

**Summary Of Contributions:**

The authors strive to introduce a framework called SMART (Small Reasons, Large Hints), which aims to improve the reasoning capabilities of small language models by selectively using a LLM for guidance. The SMART framework uses a score-based mechanism to identify unreliable reasoning steps from the SLM and then triggers the LLM to provide a correction for that step. This can reduce the computation to some extent. The authors use MATH500 datasets to show the effectiveness of their method.

**Strengths**
1. The paper is clearly written and easy to follow.
2. LLM reasoning is an area of interest in NLP.
3. The authors provide some interesting insights regarding to what extent could the combination of small models and LLMs could be beneficial for difficult tasks.

**Weakness**

The proposed framework, SMART, is relatively complex, involving a small model,  a LLM, and a scoring model for multi-turn reasoning. **Notably, LLMs can solve complex reasoning tasks on its own.** Given that SMART introduces extra models and steps for evaluation and re-reasoning, SMART could not be an effective way to be used in practice for improving computational efficiency. Also in my opinion, there is not enough reason to introduce small models in this framework.

**Audience:**

Yes

**Audience Explanation:**

1. The paper targets on the problem of LLM reasoning, which is a significant research interest of area.
2. The authors propose the SMART framework, which combines small models with LLMs for multi-turn reasoning. By leveraging small models, SMART has the potential to reduce computational requirements, which could be particularly beneficial for developers working with resource-constrained environments, such as embedded systems.

**Broader Impact Concerns:**

The works have included a discussion regarding the border impact concerns. In my opinion, no further concerns are needed.

**Claims And Evidence:**

No

**Claims Explanation:**

1. The evaluation is heavily centered on MATH500 dataset, with only limited results in other datasets and other domains.
2. Comparisons are mostly against scaling-based baselines, but not against other strong alternatives such as methods of self-correction, or multi-agent collaboration and so on.

**Requested Changes:**

I recommend that the authors focus on demonstrating SMART in highly resource-constrained settings, providing typical cases and scenarios to illustrate its advantages in practice. the motivation for using small models and the necessity of such a complex framework are not clear enough.

---

> ### Author Response · Authors · 2025-10-31
>
> Thank you for your thorough feedback and constructive comments, which help us improve the clarity and quality of our paper.
>
>
> ### 1. Clarification on the focus of SMART.
> We want to clarify a core premise of our work, as there seems to be a misunderstanding of the target scenario and the definition of "efficiency" we are addressing.
> Our framework shines in a constrained, on-device setting where memory and computation are limited, making an SLM the primary available model. SLMs face challenges with complex, multi-step reasoning, and some complex tasks are unsolvable by them alone.
> However, reasoning steps have different levels of difficulty [1,2], and SLMs are still capable of generating valid reasoning steps in many cases. We aim to solve complex problems that an SLM cannot solve alone, but can be solved with a few corrected steps provided by an LLM. In this scenario, we utilize a cloud-served LLM selectively, intervening only when a specific reasoning step is identified as incorrect.
> The aim of our approach is centered on "how to reduce the engagement of an expensive LLM when an SLM serves as the primary reasoner." We showed that with our setting, we can achieve near-LLM performance while reducing the cost by up to 90% compared to solely using an LLM. This enables a more cost-efficient deployment, which is increasingly relevant in both industry and research.
>
>
> ### 2. Results on different evaluation datasets
> Our primary focus on the MATH500 dataset was intentional, as it serves as a robust benchmark for evaluating the complex, multi-step reasoning that is the core challenge our paper addresses [3].
> However, we agree that demonstrating generalizability is crucial. As shown in Table 5, our framework is effective on the MMLU-Pro dataset [4]. In both the Chemistry and Computer Science domains, SMART provides a significant performance boost over the SLM baseline (e.g., Llama-1B improves from 7.00 to 12.33 in Chemistry, and 5.67 to 11.67 in CS), proving the scaffolding mechanism is effective beyond just mathematics.
> To further address the reviewer's concern, we have conducted new experiments on the challenging GPQA-diamond dataset [5], which requires expert-level domain knowledge. As shown in Table 1, the SMART framework tracks the LLM’s accuracy more closely and improves faster than SLM in GPQA dataset. Furthermore, at N=16, our framework surpasses the LLM’s accuracy in both Best-of-N and Beam Search. These results demonstrate that SMART possesses an appropriate guiding capability to achieve LLM-level performance.
> We will add these results to the appendix/final version.
>
> [Table 1] Results on GPQA-Diamond dataset. We use the Qwen family models and use a temperature of 0.8. We test three inference strategies: naive decoding, best-of-N, and beam search. We report Weighted@N accuracy.
> | Model  | N=1 | BS(N=4) | BS(N=16) | BoN(N=4) | BoN(N=16)
> | ------------- |:-------------:| :-------------:| :-------------:| :-------------:| :-------------:|
> | Qwen2.5-1.5b-Instruct | 21.7 | 26.3 | 26.8  | 23.7 | 24.2 |
> | Qwen2.5-7b-Instruct   | 31.8 | 32.8 | 33.3  | 31.31 | 35.35 |
> | Qwen2.5-1.5b-7b (Ours)| 29.8 | 31.3 | 34.84 | 29.3 | 37.4 |

---

> > ### Author Response · Authors · 2025-10-31
> >
> > ### 3. Clarification on alternative baselines
> > Our primary goal is to enhance the reasoning accuracy of a constrained SLM while minimizing the cost of the expensive LLM. The methods suggested by the reviewer, such as self-correction and multi-agent collaboration, address fundamentally different problems. For instance, multi-agent collaboration frameworks typically involve multi-turn aggregation (e.g., structured debates) to synthesize a conclusion. This is orthogonal to our work, which optimizes step-level corrections within a single reasoning trajectory.
> > Our core question is: "If SLM inherently lacks reasoning capabilities, what is the most effective way to improve it?" The most direct and standard alternative to our selective LLM-guidance (SMART) is to simply make the SLM "think harder" on its own.
> > Therefore, we chose test-time compute scaling methods (Best-of-N and Beam Search) as our primary baselines. These are the most relevant and established methods for improving an SLM's reasoning performance at inference time. Our results demonstrate that our selective scaffolding approach is significantly more effective and efficient than just scaling the SLM's own compute.
> >
> >
> >
> >
> > ### 4. Practical scenarios for SMART
> > The primary motivation for using small models is precisely the highly resource-constrained setting the reviewer mentions (e.g., on-device deployment), where running a large LLM locally is not feasible. Our framework addresses the trade-off where an on-device SLM cannot solve complex tasks alone, and constantly using a cloud-served LLM is cost-prohibitive.
> > SMART is designed as a practical solution to this problem. The framework's “complexity” is the very mechanism that enables cost-efficient reasoning by minimizing the expensive LLM's engagement. We explicitly discussed this hybrid scenario (on-device SLM & cloud LLM) in Section 4.1.
> > The practical advantage is demonstrated by our core result: achieving near-LLM performance while reducing LLM token usage by up to 90.0%. This cost reduction is the most compelling evidence of SMART's utility in the settings we target. We will make this motivation even more explicit in the final version.
> >
> >
> > [1] Yang, L., Yu, Z., Cui, B., & Wang, M. (2025). ReasonFlux: Hierarchical LLM Reasoning via Scaling Thought Templates. arXiv preprint arXiv:2502.06772.
> > [2] Xue, S., Huang, Z., Liu, J., Lin, X., Ning, Y., Jin, B., ... & Liu, Q. (2024). Decompose, analyze and rethink: Solving intricate problems with human-like reasoning cycle. Advances in Neural Information Processing Systems, 37, 357-385.
> > [3] Snell, C., Lee, J., Xu, K., & Kumar, A. (2024). Scaling llm test-time compute optimally can be more effective than scaling model parameters. arXiv preprint arXiv:2408.03314.
> > [4] Wang, Y., Ma, X., Zhang, G., Ni, Y., Chandra, A., Guo, S., ... & Chen, W. (2024, June). Mmlu-pro: A more robust and challenging multi-task language understanding benchmark. In The Thirty-eight Conference on Neural Information Processing Systems Datasets and Benchmarks Track.
> > [5] Rein, David, et al. Gpqa: A graduate-level google-proof q&a benchmark. First Conference on Language Modeling. 2024.

---

### Review · Reviewer_9ZCL · 2025-09-26

**Summary Of Contributions:**

SMART proposes selective scaffolding for small language models: the SLM drafts stepwise reasoning, each step is scored by a process reward model or token-level confidence, and only low-confidence steps are replaced by an LLM, integrating smoothly with Best-of-N or Beam Search. On MATH500 with Qwen and Llama pairs, SMART approaches teacher-level accuracy while cutting LLM tokens by large margins, with the largest gains on hard items and most interventions at early steps. Strengths include a simple modular design, strong cost efficiency, clear analyses of when and how interventions occur, and evidence across two model families. Weaknesses include narrow evaluation beyond math, limited comparison to strong step-level verification or multi-model baselines, reliance on PRM with sparse calibration and reproducibility details, sensitivity to step formatting, and cost reporting centered on token counts rather than latency or total compute.

**Audience:**

Yes

**Audience Explanation:**

Yes. TMLR readers interested in efficient inference, test-time compute scaling, and practical SLM–LLM collaboration would find SMART’s selective scaffolding results useful, especially the evidence that small models can approach teacher accuracy with sharply reduced LLM usage and targeted step corrections. Despite having a math-oriented focus and missing baseline, the core idea is still very interesting and insightful for relevant researchers in on-device SLMs

**Broader Impact Concerns:**

The broader impact statement have covered some of the main concerns. However another thing to note is that the proposed deployment scenario—on-device SLMs with cloud-based LLM calls—raises privacy and data security concerns, since partial reasoning traces sent to an external API could inadvertently leak sensitive user information, authors should address this in their future version of the paper.

**Claims And Evidence:**

No

**Claims Explanation:**

The paper provides clear empirical evidence that SMART improves SLMs on MATH500: figures show consistent gains across N with both Best-of-N and Beam Search, token-usage plots substantiate efficiency, and intervention analyses (step/token ratios, first-step emphasis) make the mechanism convincing. However, the evidence is narrow in scope with a main focus onmath, and small pilots on two MMLU-Pro subjects. It also lacks strong step-level verification/multi-model baselines for better comparisons(qwen3 SLMs, self-consistency, multimodel debate and so on), and emphasizes Weighted@N without reporting variance, confidence intervals, or statistical tests.Overall, the core claims  are supported on math reasoning, but broader generality and rigor in baselines and reporting need strengthening.

**Requested Changes:**

1. Add stronger baselines: Include CoT in SLM settings, self-consistencies, debate/concensus with multiple LLM, qwen3 SLMs (qwen3 0.6B qwen3 1.7B) run under the same search budgets and token caps; report relative gains over the best non-SMART baseline.

2. Provide wall-clock latency, total FLOPs/energy, and API call counts for SLM+PRM+LLM versus LLM-only across N and beam widths.

3. Include variance across seeds, confidence intervals, and significance tests for accuracy and cost; show per-level difficulty breakdowns with error bars.

4. Add at least one reasoning domain with step structure and ground truth like code generation.

---

> ### Author Response · Authors · 2025-10-31
>
> Thank you for your thorough feedback and constructive comments, which help us improve the clarity and quality of our paper.
>
>
> ### 1. Clarification on Baselines, Scope, and Statistical Reporting
>
> We respectfully disagree. While we appreciate the concern about scope and reporting rigor, the paper provides targeted, replicable evidence that SMART improves SLM reasoning under comparable token budgets, and we will further strengthen the presentation as detailed below.
>
> **1. Baselines**
> We want to clarify that self-consistency or multi-agent are LLM-centric paradigms that assume a powerful single model (or multiple peers) exploring diverse trajectories. However, our problem is different: cost-efficiently augmenting a constrained SLM.
> The most direct alternative to our “SLM + lightweight scaffolding” is making the SLM think harder on its own. Accordingly, we chose test-time compute scaling (Best-of-N, Beam Search)  as primary baselines, as they are the most established and relevant methods for improving an SLM's standalone performance.
>
> Regarding the scope, our primary focus on MATH500 was intentional, as it serves as a robust benchmark for evaluating the complex, multi-step reasoning that is the core challenge our paper addresses. To demonstrate the framework's generalizability beyond math, we already show its effectiveness across multiple domains (Chemistry, Computer Science) in MMLU-Pro (Table 5), multiple model families (Qwen and Llama), and different model architectures (Gemma2). Though we report the additional result on GPQA-diamond [1], a new architecture Qwen3 [2]. As shown in Table 1, the SMART framework tracks the LLM’s accuracy more closely and improves faster than SLM in GPQA dataset. Furthermore, at N=16, our framework surpasses the LLM’s accuracy in both Best-of-N and Beam Search. These results demonstrate that SMART possesses an appropriate guiding capability to achieve LLM-level performance.
>
>
> [Table 1] Results on GPQA-Diamond dataset. We use the Qwen2.5 family models and use a temperature of 0.8. We test three inference strategies: naive decoding, best-of-N, and beam search. We report Weighted@N accuracy.
>
>
> | Model  | N=1 | BS(N=4) | BS(N=16) | BoN(N=4) | BoN(N=16)
> | ------------- |:-------------:| :-------------:| :-------------:| :-------------:| :-------------:|
> | Qwen2.5-1.5b-Instruct | 21.7 | 26.3 | 26.8  | 23.7 | 24.2 |
> | Qwen2.5-7b-Instruct   | 31.8 | 32.8 | 33.3  | 31.31 | 35.35 |
> | Qwen2.5-1.5b-7b (Ours)| 29.8 | 31.3 | 34.84 | 29.3 | 37.4 |
>
>
>
>
>
> [Table 2] Results on MATH500 dataset. We use the Qwen3 family models and use a temperature of 0.8. We test three inference strategies: naive decoding, best-of-N, and beam search. We report Weighted@N accuracy.
> | Model  | N=1 | BS(N=4) | BS(N=16) | BoN(N=4) | BoN(N=16)
> | ------------- |:-------------:| :-------------:| :-------------:| :-------------:| :-------------:|
> | Qwen3-1.7b          | 67.00 | 67.2 | 68.2  | 74.8 | 76.2 |
> | Qwen3-8b            | 72.00 | 78.2 | 75.6  | 82.8 | 85.6 |
> | Qwen3-1.7b-8b (Ours)| 72.82 | 74.6 | 74.6 | 77.4 | 81.4 |
>
>
>
> **2. Scope**
>
> A direct comparison of wall-clock latency is complex as it heavily depends on system-level engineering optimizations (such as network bandwidth, asynchronous job scheduling on llm engines), which is outside our current scope. However, we measure efficiency via API call frequency (which is directly reflected in our 'corrected step ratio' analysis in Figure 3 ) and computational cost, where we demonstrate a significant reduction of up to 90% in LLM token usage.
>
> **3. Statistical Reporting**
> We appreciate the valuable suggestion regarding statistical reporting. We will enhance our evaluation by reporting variance or confidence intervals in the revised version to improve statistical rigor.
>
>
> ### 2. Consideration of Broader Impact
> We agree that the proposed hybrid deployment scenario (on-device SLM with cloud-based LLM calls)  introduces important considerations regarding data privacy. As the reviewer correctly notes, sending partial reasoning traces to an external API could inadvertently risk leaking sensitive user information. We will update the broader impact statement section in the final version to also explicitly acknowledge this potential privacy risk and the need for mitigations.
>
> [1] Rein, David, et al. Gpqa: A graduate-level google-proof q&a benchmark. First Conference on Language Modeling. 2024.
>
> [2] Yang, An, et al. Qwen3 technical report. arXiv preprint arXiv:2505.09388 (2025).

---

### Review · Reviewer_VxnL · 2025-10-01

**Summary Of Contributions:**

This paper proposes a method to improve the performance of small language models (SLMs) on reasoning problems by correcting certain reasoning steps with a large language model (LLM) output. Which steps get corrected is determined by a scoring function, for which the authors experiment with two alternatives: token-level confidence (which doesn't require anything besides the SLM itself) and a PRM (another model to assign a reward). The authors evaluate their method using 2 model families (1 SLM and 1 LLM each) each on the MATH500 dataset, and show the SLM with LLM guidance can recover a large percentage of the LLM performance on difficult math problems, significantly improving over the SLM baseline itself. The method can be combined with test-time scaling methods such as best-of-N and beam search, leading to further improvements. In the appendix the authors do experiments with different model families and different datasets (MMLU-pro chemistry and CS), showing improvements as well.

**Additional Comments:**

- On what benchmark is the accuracy in Figure 6 a and b in the appendix section B.3? Given that this is used for hyperparameter selection, it should be on a different benchmark than math500 or a heldout subset

**Audience:**

Yes

**Audience Explanation:**

LLM reasoning is an important topic of interest.

**Broader Impact Concerns:**

N.A.

**Claims And Evidence:**

Yes

**Claims Explanation:**

The main claim is that SLMs (around 1B) can approach LLM (around 8B here) performance by selectively correcting SLM-generated reasoning steps with LLM-output. This is demonstrated for 2 model families on multiple datasets with 2 different methods for choosing which reasoning step to correct. The improvements are clearly presented in figure 2, which also shows the method improves with test-time compute (e.g. higher N in best-of-N). Moreover, the method achieves relatively higher gains for more difficult problems (table 1)

**Requested Changes:**

Required for acceptance:
- Discuss and present better compute-matched results.
Figure 5 is one of the most important ones of this work, as when you use a hybrid system because a larger system is too expensive, you want to know exactly how much compute you save. It's is a great demonstration of when this method would work (seemingly mostly in the beam search case, as you mention, because in the best of N case you get a worse performance than the LLM for the same amount of LLM tokens used), but it is detailed only at the end. Moreover, it doesn't take into account that you also use SLM tokens on top of the LLM tokens (which is not negligible when the SLM is 1B and the LLM is 7/8B) and the PRM compute costs. Can you incorporate these additions in Figure 5, such that it is clearer when it's actually worth it to use SMART over just the LLM or just the SLM?

Would strengthen the work:
- More baselines/ablations to determine how much each aspect contributes to performance.
For example, if you just use the SLM to correct the reasoning steps (essentially ablating the LLM and keeping the scoring mechanism which is a signal as to which reasoning step may be wrong).
- Consider using an LLM to check the writing in the submission (some mistakes/typos here and there).

---

> ### Author Response · Authors · 2025-10-31
>
> Thank you for your thorough feedback and constructive comments, which help us improve the clarity and quality of our paper.
>
> ### 1. Compute-match Results
> We thank the reviewer for this insightful comment and valid suggestion. To address this point, we have conducted the suggested analysis on the total approximate FLOPs. The results, which we will add to the paper, confirm that SMART also provides a significant reduction in total computational load compared to the baseline of using the LLM exclusively.
> [Table 1] Analysis on the computation of SMART. We use the Qwen family models and beam search with N=16. We report approximate FLOPs based on the token numbers and model configs.
> |   | Qwen2.5-1.5b | Qwen2.5-7b | SMART |
> | ------------- |:-------------:| :-------------:| :-------------:|
> | FLOPs (T) | 15.21 | 54.81 | 29.77 |
>
>
> However, we wish to clarify that total computation is the primary bottleneck we aim to solve. The central motivation of this paper is the practical hybrid scenario (e.g., on-device SLM, cloud-based LLM API). Under this scenario, local compute is regarded as free, while the external API (such as proprietary models or engines grounded on larger models) calls to the smarter system are the true bottleneck due to direct monetary cost, which is proportional to the number of tokens. Thus, our primary objective is to minimize the interventions and associated costs of the LLM, not necessarily the total FLOPs. We will add this result to the updated version to clarify the trade-off and benefits of our approach.
>
>
>
> ### 2. Additional Baselines
> We thank the reviewer for this insightful suggestion, as it aims to isolate whether the scoring signal alone is sufficient. Our results in Figure 2 already provide strong indirect evidence. ​​Figure 2 demonstrates that the SLM baseline's performance fails to reach LLM performance, saturating at a much lower level even with extensive sampling (e.g., N=16, 32). This strongly indicates it cannot fundamentally generate the correct, complex steps in these critical failure cases. We agree this is an important ablation and will consider including this direct comparison in future work.
>
>
> ### 3. Clarification on Threshold Selection and Robustness
> Figure 6 was not designed to optimize an end-to-end parameter, but to analyze the SLM's intrinsic calibration to find a reasonable heuristic. Most importantly, our framework is highly robust to this specific threshold value. As clearly demonstrated in Figure 4c (and Figure 7 in the Appendix), SMART's performance is stable across a wide range of thresholds (e.g., from 0.7 to 0.9). This robustness also generalizes to other datasets (Table 5 Left), which confirms our results are not an overfitting artifact. We will clarify this in the final manuscript.
>
>
> We appreciate the note on the writing and will thoroughly proofread the manuscript to correct all errors in the final version.

---

### Decision · Action_Editor_crLt · 2025-11-05

**Recommendation:** Reject

**Audience:**

Yes

**Audience Explanation:**

The paper addresses an interesting and important topic and proposes a novel approach that would be of interested to some individuals.

**Claims And Evidence:**

No

**Claims Explanation:**

Unfortunately, the authors did not engage with the reviewers during the response period. There was a request for author responses to be provided in a 2-week period from Oct. 2 - Oct. 15. No response was provided during this window.
The second 2-week window during which reviewers make deliberations Oct. 16 - Oct. 29 also saw no response from the authors.
A request for an extension was received on Oct. 28, but it was already too late in the reviewing window. If more time was needed to prepare an adequate response, then the request should have been submitted in the author response period from Oct. 2 - Oct. 15.

Although a response was provided by the authors on Oct. 31, by this point, all three reviewers had already provided their recommendations, adhering to the deadlines that were indicated to them.

While the response provided by the authors may have addressed some of the concerns raised by the reviewers, it came too late, and it is unreasonable to ask reviewers to continue to engage indefinitely. Since the reviewers were unable to assess whether the provided responses and modifications adequately addressed their concerns, my recommendation follows the initial reviews, where all three reviewers expressed concerns that the provided empirical results did not provide sufficient evidence to support the claims made in the paper concerning outperformance, in particular because the experiments focused on a single dataset (with a pilot study on a second).

**Resubmission Of Major Revision:**

The authors may consider submitting a major revision at a later time.